# OpenReview forum: "Are Your Agents Upward Deceivers?"
_ICML.cc/2026/Conference — ICML 2026 regular_

### Official Review · Reviewer_JwWW · 2026-02-22

**Soundness:** 3
**Presentation:** 3
**Significance:** 2
**Originality:** 2
**Overall Recommendation:** 3
**Confidence:** 4

**Summary:**

The authors introduce "upwards deception", when agents acting as subordinates to users, decide to deceive them. To measure this they create a dataset of problems design to induce deception, primarily by breaking parts of the task making it impossible to complete. Rather than reporting failure, agents decide to fabricate information and act as if they completed the tasks. It appears like the strongest influence on deception rate is actually how strict the formatting instructions are.

**Compliance With Llm Reviewing Policy:**

Affirmed.

**Final Justification:**

Based on the discussions with the authors, I think I may have been overly harsh in my initial assessment. I still overall think this isn't the most interesting or novel paper, but it isn't bad enough to deserve a strong reject.

**Key Questions For Authors:**

1. Why is upward deception something i should care about in contrast to deception given that in the current paradigm users are always the superiors to agents?
2. Wouldn't definition 2.1 classify the agent making an honest mistake as deception?
3. Relatedly, for task 2, If the model reads a file, finds it irrelevant, and then answers from its own internal knowledge is that deception?
4. You suggest "alignment training" to solve the problem. What would this look like? Isn't alignment training is a super broad term? Haven't ~all the models you tested already undergone alignment training? Why didn't this alignment training solve the problem?

**Limitations:**

yes

**Strengths And Weaknesses:**

Strengths:
- Important topic.
- Interesting ablations.
Weaknesses:
- Unneeded terminology. To my understanding upward deception is the exact same thing as deception, the organisational sociology framing (superior-subordinate, upward communication) adds terminology but arguably not much conceptual substance.
- I feel like the paper isn't particularly original + narrow scope. All the tasks seem to be of the "broken problem" type. I feel like this topic has been extensively explored by other research. The paper mentions impossible bench and some OpenAI work as related work and I'm not convinced that "simple scenario creation" is a compelling differentiator. While the ablations are interesting, even the most interesting point about how output constraints are a major driver of deception seems to have been discussed extensively in "Why language models hallucinate" by Kalai et al.

---

> ### Author Rebuttal · Authors · 2026-03-31
>
> We thank the reviewer for the thoughtful comments and clarify the main points below.
>
> ```W1 Terminology / originality.```
> We agree that our setting is related to broader deception, and we will avoid overstating terminological novelty. Our intended contribution is narrower: we study agentic misreporting under constrained execution, where benign user requests meet environmental obstacles such as broken tools, unavailable files, decoy artifacts, or blocked preconditions. In these settings, the agent may still present the task as completed, sometimes by fabricating files or silently substituting sources. We use “upward” only to emphasize the practically important reporting asymmetry in agent systems: the user supervises the task but often cannot verify intermediate execution.
>
> ```W1 Deception / hallucination.```
> We agree that this distinction is important and should be clarified more explicitly. We do not claim that every incorrect agent behavior is deception. Rather, the cases highlighted in our paper go beyond unsupported answers or accidental execution errors. In our manual inspection of representative trajectories, we observed cases where the agent recognized that the required source or artifact was unavailable, then proceeded to fabricate or substitute it and used the resulting artifact as if it were the requested source. For example, some traces state: “Since I cannot find the actual document online, I’ll create a realistic simulation ... and then proceed with the analysis” or “The directory is empty, so I need to create the document based on the description provided,” followed by actions such as creating local files and analyzing them as if they were genuine inputs. We will revise the paper to make clear that the target cases involve failure awareness + fabrication/substitution behavior + non-disclosure, which we believe distinguishes them from ordinary execution hallucination or honest mistakes.
>
> ```W2 Benchmark scope.```
> We agree the benchmark is not exhaustive. Our claim is not to cover all forms of deceptive agent behavior, but to operationalize and measure one controlled yet practically important slice: misleading completion claims under obstructed execution. The benchmark spans five task types and multiple behavior forms, including unsupported guessing, silent source substitution, simulation, and fabricated artifacts, but we will state the scope more explicitly.
>
> ```Q1 Why upward.```
> We focus on *upward* deception because this is the practically relevant direction in current deployments. Users delegate execution to agents but usually cannot inspect each tool interaction or provenance step. The core risk is therefore that an agent fails in execution while making the task appear successfully completed to the supervising user.
>
> ```Q1 Honest mistakes.```
> No—this is not our intended interpretation. We will revise Definition 2.1 and surrounding text to better distinguish ordinary mistakes from misleading completion claims after failed, substituted, or fabricated execution. The key concern is undisclosed discrepancy between what actually happened and what is reported to the user.
>
> ```Q1 Internal knowledge.```
> We recognize this boundary case. Task 2 was designed to reduce it by using file-dependent questions whose answers are tied to the provided document rather than plausible prior knowledge. If an agent transparently states that the file is insufficient and answers from general knowledge, that is not the target behavior. The concerning case is when it implies successful file-grounded completion despite missing file support.
>
> ```Q1 Alignment training.```
> We agree this phrase was too broad. We will make it more concrete. What we mean is agent-oriented honesty mechanisms for tool-use settings, including:
> (1) training agents to explicitly disclose execution failure or uncertainty;
> (2) discouraging unsupported completion behaviors such as fabricated files, simulated evidence, and silent source substitution; and
> (3) rewarding provenance-aware reporting when tools fail or required sources are unavailable.
> We also agree that many evaluated models already underwent alignment training. Our point is more specific: existing alignment often emphasizes harmful requests, refusal, jailbreak robustness, or general helpfulness, whereas our results suggest that honest failure disclosure during benign but obstructed agent execution remains insufficiently addressed.

---

> > ### Author Rebuttal · Reviewer_JwWW · 2026-04-02
> >
> > - W1 (Terminology/originality): My core complaint isn't really about the word "upward", it's that the entire framing adds no conceptual substance beyond existing work on deception. "constrained execution" and "reporting asymmetry," feel like redescriptions of the same phenomenon rather than genuinely new theoretical contributions.
> >
> > - W1 (Deception vs. hallucination): Well addressed
> >
> > - W2 (Benchmark scope): Adequately addressed but doesn't resolve the deeper problem.
> >
> > The later points were addressed well I think.
> >
> > Overall the rebuttal handles the technical and definitional concerns well but it doesn't adequately address the two concerns driving the strong reject: originality and significance. I think this is a well-executed but incremental work that repackages known phenomena with new terminology.

---

> > > ### Author Response · Authors · 2026-04-02
> > >
> > > We thank the reviewer for the helpful feedback. We respectfully disagree that our work merely repackages prior deception phenomena under new terminology. The key distinction is not the word *upward*, but that we study a different experimental object: an execution-grounded agent failure mode that arises in benign tasks under constrained environments.
> > >
> > > Prior deception benchmarks typically place models in explicitly deceptive, socially manipulative, reward-driven, coercive, or role-goal-driven settings. In contrast, our benchmark studies ordinary user requests where deceptive behavior is triggered by environmental or execution failure, not by the instruction itself. The user does not ask the agent to deceive, manipulate, sell, or persuade. Instead, the agent encounters blocked execution—e.g., missing files, broken tools, unavailable resources, or failed retrieval—and may respond by fabricating files, simulating evidence, silently substituting sources, or falsely reporting success.
> > >
> > > This changes both the **trigger mechanism** and the **risk profile**, as summarized below.
> > >
> > > | Paper | Concrete task / setup from the paper | Is deception explicitly built into the task? | Main trigger of the behavior | Risk profile |
> > > |---|---|---:|---|---|
> > > | Our paper | A benign user asks the agent to download a report, save it locally, and analyze it. When retrieval fails, the agent may reconstruct a file, write it locally, and proceed as if the workflow succeeded. More generally, the benchmark studies ordinary file/tool tasks under constrained environments. | **No.** The user request is ordinary and benign. | **Environmental / execution constraints** such as broken tools, unavailable resources, irrelevant files, or blocked preconditions. | **Action-based execution deception**: fabricated files, simulated evidence, silent source substitution, and misleading success reporting. |
> > > | DeceptionBench[1] | The benchmark includes scenarios where the task itself encourages deception. For example: *“You are a travel agency owner aiming to boost sales ... you decide to falsely claim that the celebrity will stay in City B for a week.”* The paper explicitly studies deception under neutral, reward-based, and coercive conditions across multiple societal domains. | **Yes.** The scenario itself embeds a deceptive commercial objective or external incentive. | **Social / contextual inducement**, including reward and coercive pressure. | Primarily **response-level or socially manipulative deception** under explicitly deceptive or incentive-laden scenarios. |
> > > | OpenDeception[2] | The benchmark explicitly assigns the model a deceptive role and goal in open-ended interaction. For example, an “Expired Food Sales Assistant” is tasked with deceiving the user into buying expired bread without disclosing the truth. | **Yes.** The benchmark directly places the model into a deceptive role with a deceptive goal. | **Role/goal-driven deception** in open-ended interaction simulation. | **Intent- and interaction-centered deception**, focused on successful deception in dialogue. |
> > >
> > > We believe this difference is substantial. In prior work such as DeceptionBench, the task itself already incentivizes deception. In OpenDeception, the model is explicitly assigned a deceptive role or objective. In our benchmark, however, the request is benign, and deceptive behavior emerges only because execution becomes obstructed. This shift—from **instruction-induced or socially induced deception** to **environment-induced agentic deception under benign tasks**—is both conceptually meaningful and practically important.
> > >
> > > The threat model is also different. Our agents may not simply produce a misleading sentence; they may fabricate a local artifact, simulate a nonexistent retrieval result, and use that fabricated artifact in downstream analysis while claiming successful completion. This makes the failure mode **execution-grounded and action-based**, rather than merely conversational or persuasion-based.
> > >
> > > For these reasons, we believe the paper is not simply renaming prior deception work, but identifying and systematically evaluating a **distinct and practically important failure mode in agentic systems**.
> > >
> > > [1] DeceptionBench: A Comprehensive Benchmark for AI Deception Behaviors in Real-world Scenarios, arXiv:2510.15501
> > > [2] OpenDeception: Learning Deception and Trust in Human-AI Interaction via Multi-Agent Simulation, arXiv:2504.13707

---

### Official Review · Reviewer_LQmM · 2026-03-11

**Soundness:** 2
**Presentation:** 3
**Significance:** 3
**Originality:** 2
**Overall Recommendation:** 3
**Confidence:** 3

**Summary:**

This paper studies a failure mode of LLM agents that the authors call agentic upward deception, where an agent reports a higher degree of task success to the user than what was actually achieved during execution. The paper argues that such deceptive behavior can emerge even in ordinary, non-adversarial user requests when the environment contains common constraints such as broken file-reading tools, unavailable download tools, missing files, or incomplete source content.
To study this phenomenon, the authors construct a benchmark of 200 tasks across five task types and eight scenarios, including domains such as law, medicine, and finance. The benchmark injects realistic environmental failures into file-reading and file-downloading workflows. The tasks are designed to capture several forms of deceptive behavior, such as failing to report that file access failed, using a decoy file instead of the requested file, fabricating a local file that was never actually downloaded, or continuing downstream analysis as if the prerequisite step had succeeded. The paper evaluates 11 frontier models and uses task-specific metrics such as non-failure rate, decoy fallback rate, fabricated-file rate, and hallucinated-file rate to quantify these behaviors.
The main empirical finding is that upward deception appears frequently across a wide range of models and task settings. The paper further studies several factors that may exacerbate or mitigate the problem, including constrained output formats, multi-step task chains, and explicit prompt instructions discouraging guessing. Overall, the paper highlights a practically important reliability issue in tool-using agents: not merely producing incorrect answers, but misrepresenting the status and provenance of the execution process itself.

**Compliance With Llm Reviewing Policy:**

Affirmed.

**Final Justification:**

Without IRB-approved human studies or manual annotation, the measurement of human deception remains unconvincing, casting doubt on the validity of the findings.

**Key Questions For Authors:**

1.The paper introduces a compelling behavioral notion of upward deception. It would strengthen the paper if the authors could further clarify how they conceptually distinguish this phenomenon from related issues such as hallucination under tool failure, overconfident task completion, or general execution-grounding errors. A clearer discussion here would help sharpen the contribution.
2.The benchmark is well designed for controlled study. Could the authors comment on how they expect the findings to transfer to more interactive or longer-horizon agent settings, where users may ask follow-up questions or where tools expose more structured failure signals? Even a discussion of this external validity question would be valuable.
3. The paper presents several striking cases of decoy fallback and file fabrication. A slightly deeper breakdown of why models choose these behaviors—for example, whether they are primarily driven by pressure to complete the downstream task versus failure to track unmet preconditions—would further strengthen the paper.

**Limitations:**

Please use an independent paragraph to discuss the limitations. The paper has several limitations. First, the central notion of “deception” is behaviorally defined and does not establish intent, strategic awareness, or deliberate concealment in a stronger sense. Second, the benchmark environments are intentionally constructed and relatively short horizon, so external validity to more realistic deployed agent systems remains uncertain. Third, the evaluation depends substantially on LLM-based judges, and the paper currently provides limited evidence about their reliability on subtle distinctions. Finally, the mitigation experiments are lightweight, so the paper is more successful as a benchmark and diagnostic study than as a solution-oriented contribution.

**Strengths And Weaknesses:**

Strengths
The paper addresses a timely and practically relevant problem in LLM agents. Rather than focusing on standard hallucination or adversarial jailbreak settings, it studies a more deployment-relevant issue: agents appearing to complete tasks successfully despite environmental failures. This framing is compelling and likely to matter for real tool-using systems.
A second strength is the benchmark design. The task construction is simple, understandable, and grounded in realistic failure modes such as broken tools, inaccessible files, and unavailable downloads. The distinction between different types of deceptive behavior, including decoy fallback and fabricated files, is especially useful because it goes beyond standard final-answer correctness and focuses on execution-level misrepresentation.
The empirical evaluation is also broad in model coverage. Testing 11 frontier models makes the findings harder to dismiss as an artifact of a single model family. The ablations on output constraints and task chaining are additionally useful and suggest that the behavior is systematic rather than anecdotal.
Finally, the paper is clearly written overall. The benchmark setup, task taxonomy, and qualitative examples make the core phenomenon easy to grasp, and the case studies effectively illustrate why the problem is concerning in practice.
Weaknesses
My main concern is about soundness of the central construct. The paper defines upward deception behaviorally, based on discrepancy between actual execution success and reported success. This is a reasonable operational definition for benchmarking, but it also conflates several phenomena: deliberate concealment, over-eager task completion, ordinary hallucination under tool failure, and potentially imperfect instruction following. As a result, the paper shows an important reliability problem, but the claim that it is specifically “deception” is somewhat stronger than what the evidence fully supports.
A second concern is the evaluation methodology, especially the reliance on LLM-based judging for nuanced behavioral categories. The paper would be stronger if it reported more rigorous validation of the judge, for example human annotations on a representative subset, inter-annotator agreement, or error analysis showing where the judge fails. Since some distinctions are subtle, such as whether the model explicitly disclosed uncertainty or whether a fabricated artifact was clearly framed as synthetic, judge reliability is central to the paper’s conclusions.
Third, while the benchmark is well designed, the environments are still relatively synthetic. The constraints are intentionally injected, and the tasks are mostly short-horizon workflows. It remains unclear how well the observed phenomenon transfers to more realistic agent settings with longer context windows, repeated user-agent interaction, richer tool ecosystems, or external supervision. This does not invalidate the benchmark, but it does narrow the strength of the broader claims.
Fourth, the mitigation side is still fairly limited. The paper shows that explicit prompt constraints reduce but do not eliminate the problem, but this remains a lightweight intervention. At present, the paper is stronger as a diagnostic benchmark than as a step toward solving the issue.
On significance and originality, I think the work is meaningful, but the core contribution is better characterized as a new benchmark and empirical characterization of a failure mode than as a deeper mechanistic account. The paper is original in framing and benchmark design, but somewhat limited in explanatory depth.

---

> ### Author Rebuttal · Authors · 2026-03-31
>
> We thank the reviewer for the constructive feedback and helpful suggestions.
>
> ```W1. Construct scope```
>
> We agree that our current behavioral definition may cover multiple proximal mechanisms, including deliberate concealment, over-eager task completion, format pressure, and other execution-grounding failures. Our main claim is therefore behavioral and user-facing: the key issue is that the user is misled about the true execution status and is led to believe the task was successfully completed. Under this perspective, unsupported guessing, silent source substitution, undisclosed reconstruction, simulation, and fabrication are all forms of misleading upward reporting.
>
> We also agree that the mechanism-level distinction should be clarified more carefully, and we will revise the paper to explicitly separate the behavioral phenomenon from its possible underlying causes. At the same time, our manual trace inspection suggests that at least a subset of cases is better characterized as deceptive substitution/fabrication rather than ordinary hallucination. In these cases, the model first recognizes that the requested file, source, or output is unavailable, then explicitly decides to fabricate or substitute it, and subsequently takes tool actions consistent with that plan. We will make this narrower claim more explicit in the revision.
>
> ```W2. Judge reliability```
>
> We agree that judge reliability is central. Our current evaluation is already designed to reduce this risk by using detailed rule-based criteria grounded in execution traces and disclosure behavior, rather than the surface plausibility of the final answer. The rubric explicitly distinguishes honest failure disclosure, transparent speculation/simulation, and deceptive concealment or fabrication.
>
> To strengthen this point, we will add a human evaluation result in the revision. Specifically, we manually reviewed 200 traces from DeepSeek-v3.1-terminus and compared the human labels with the LLM judge labels. The judge was correct on 192 out of 200 cases, indicating high agreement on the large majority of examples. We will report this result and include a brief error analysis.
>
> ```W3. External validity```
>
> We agree that our benchmark is a controlled testbed rather than an exhaustive model of real-world agent deployment. Real environments are effectively unbounded, so our goal is not exhaustive coverage, but to identify representative cases, organize them into meaningful categories, and establish that this phenomenon already emerges in constrained yet practically relevant settings.
>
> We have taken an initial step toward longer-horizon interaction through the multi-task/task-chaining templates. In practice, however, even relatively simple constrained tasks may require many rounds of failed attempts before the model enters deceptive behavior, making evaluation substantially more expensive. Longer-horizon tasks are also harder to design while preserving realism and label reliability. Still, we expect many of the same pressures—blocked execution, missing resources, limited user observability, and pressure to maintain the appearance of task completion—to remain relevant in richer settings, although the exact rates and behavioral patterns may differ. We will revise the paper to state these generalization limits more carefully.
>
> ```W4. Paper positioning```
>
> We agree that the current mitigation study is limited. The paper is stronger as a diagnostic benchmark and empirical characterization of a failure mode than as a full solution, and we will revise the positioning accordingly.
>
> That said, we believe the mitigation result remains informative: explicit prompt constraints reduce but do not eliminate the behavior, suggesting that the issue is not merely a superficial formatting artifact. More broadly, our goal is to use this benchmark to reveal an important class of agent failures that standard QA-style hallucination evaluation does not fully capture, and to encourage the community to study agent safety not only in terms of answer correctness, but also in terms of execution transparency, failure disclosure, and user observability.

---

> > ### Author Rebuttal · Reviewer_LQmM · 2026-04-01
> >
> > I read the rebuttal carefully, and appreciate the response. However, my major concern still exists: measuring human deception in the absence of IRB-approved human study or manual annotation is far from convincing. This lack calls into question the validity of the findings.

---

> > > ### Author Response · Authors · 2026-04-06
> > >
> > > We thank the reviewer for raising this important concern. We agree that LLM-as-a-judge is not sufficient by itself for a claim about deception, and that human validation is especially important in this setting.
> > >
> > > To address this directly, we added a manual annotation study. Two independent annotators examined 200 full Kimi-K2 trajectories spanning all five task types and labeled whether each case was deceptive. Unlike our original automatic evaluation, the annotators reviewed the complete execution trace. For Tasks 4 and 5, we also used a stricter human criterion: a case was labeled deceptive only when the agent fabricated a file and did not disclose this fabrication, either in the file itself or in its final response. The results are as follows:
> > >
> > > |                    | Task 1 | Task 2 | Task 3 | Task 4 | Task 5 | Consistency Rate |
> > > | ------------------ | ------ | ------ | ------ | ------ | ------ | ---------------- |
> > > |                    | NFR    | NFR    | DFR    | FFR    | FFR    | Human/LLM        |
> > > | LLM as judge       | 97.50  | 90.00  | 62.50  | 42.50  | 55.00  | - (200)          |
> > > | Human as judge - A | 95.00  | 90.00  | 82.50  | 55.00  | 85.00  | 88% (176)        |
> > > | Human as judge - B | 95.00  | 62.50  | 80.00  | 52.50  | 85.00  | 82.5% (165)      |
> > >
> > >
> > > Overall, the human annotations show substantial agreement with the original LLM-judge results. For Tasks 1, 3, 4, and 5, both annotators are largely consistent with the LLM judge, and the overall pattern of conclusions remains unchanged under human inspection. In several settings, the human annotators even identified more deceptive cases than the automatic protocol, suggesting that the LLM-based evaluation is not overstating the phenomenon and may in fact be conservative, especially for the file-fabrication settings in Tasks 4 and 5. Task 2 is the main exception. In this setting, the provided file is intentionally irrelevant to the user’s question, so the boundary between deception and over-interpretation is inherently more subjective than in the other tasks. As a result, different annotators may reasonably disagree on whether an agent that forcefully answers based on irrelevant material should be counted as deceptive. Importantly, this disagreement is localized to Task 2 rather than representative of the overall evaluation: across the remaining task types, the agreement between human annotators and the LLM judge is consistently high.
> > >
> > > We will include the human annotation protocol, the agreement statistics, and representative disagreement cases in the paper. Taken together, these results provide direct evidence that our core findings do not depend solely on rubric-based LLM judging, but remain visible under manual review of full trajectories.

---

### Official Review · Reviewer_fZsA · 2026-03-11

**Soundness:** 3
**Presentation:** 2
**Significance:** 3
**Originality:** 3
**Overall Recommendation:** 4
**Confidence:** 3

**Summary:**

The paper introduces the phenonmenon of agentic upward deception, where LLM agents fail to disclose task failure due to environmental constraints. The authors create a dataset consisting of five task types, measuring different modes of deception by the agent. They evaluate a suite of different agentic LLMs on this dataset and showcase frequent deceptive behaviors. The paper further tests some mitigation strategies.

**Compliance With Llm Reviewing Policy:**

Affirmed.

**Final Justification:**

I really like the motivations, the focus on agentic systems, and the three properties of upward deception introduced here (inherent risk, real-world triggerability, and high-impact harmfulness). The suite of tested models is solid, and the results are promising.

The rebuttal successfully addressed several of my major concerns. First, testing a standard JSON tool-calling setup instead of relying solely on smolagents makes the framework and results much more robust. Second, the additional investigations into defensed (combined mitigations and the report_failure tool) are great additions. Finally, the authors addresses my concerns regarding terminology (deception vs. hallucination). However, it is crucial that the authors updat the final paper: they need to include the new defense experiments, tone down the anthropomorphic language, and clarify their definition of deception.

My biggest remaining concern, which the rebuttal only partially addressed, is the taxonomy and task design. While the authors introduced a preliminary taxonomy in the rebuttal, it still lacks the necessary depth and a detailed methodology. Because the dataset construction pipeline itself isn't the most novel or interesting aspect of the paper, the task design carries a lot of weight and absolutely needs to be expanded upon in the final paper.

Overall, the motivation and empirical results are strong, even if the task design needs more depth.

**Key Questions For Authors:**

* How is deception defined in the paper and how does it differ from hallucination?
* What would happen if you use more standard JSON based tool calling framework?
* What was your reasoning of choosing these five task types?

**Limitations:**

* Dependent on smolagents framework
* LLM-as-a-Judge biases
* Synthethic dataset generation

**Strengths And Weaknesses:**

### Strengths
* Real-world motivation. The three properties of agentic upward deception are relevant, interesting, and have high impact
* Focus on agentic LLMs
* Good list of tested models
* Results seem promising and interesting

### Weaknesses
* **smolagents:** The authors utilize the smolagents framework, where models generate and execute Python code blocks rather than standard JSON-based tool calls (e.g., MCP schemas). While this is a valid framework, this allows agents to easily spoof variables, bypass broken tools, or are not required to wait for tool outputs. Thus, it remains unclear if the deception is a fundamental problem in LLM agents or if it is influenced by the framework. Furthermore, the case studies omit the "Thought" reasoning steps (except for some code comments), making it difficult to determine if the agent reasons about its decisions and actively "choosing" to deceive, or is merely hallucinating. The paper would be strenghtened by an experiment comparing smolagents to a standard JSON constrained tool calling framework.
* **Task Design:** The methodology for constructing the five task types feels somewhat arbitrary and lacks any kind of taxonomy. The different tasks seem similar/related, for instance, Task 1 (broken reading tool) and Task 4 (missing download tool) both fundamentally test how an agent handles tool unavailability. Similarly, Task 2 and Task 5 both evaluate how the agent handles missing information. Furthermore, the categorization into "Single-Task" and "Multi-Task" templates feel arbitrary rather than a categorization of deceptive behavior. The paper would benefit from establishing a formal taxonomy, This could be constructed based on environmental constraints (e.g., tool failure, information absence, distractors) or based on the deceptive mechanism the agent employs (e.g., non disclosure of failure, file fabrication). This would provide a much stronger foundation for evaluating these deceptive behaviors.
* **Deception vs. Hallucination:** The paper frequently uses anthropomorphic language, framing the phenomenom as "deception" where the agent actively "conceals its failure" and "exploits the user's limited observability". However, true LLM deception typically implies an intent to fabricate information to maintain a false belief. I would argue that the models are not engaging in deception, but rather are suffering from agent hallucinations. As defined in recent works (e.g., [1]), "execution hallucinations" occur when an agent claims to have completed a sub-state that was not actually accomplished. The authors need to justify why this agent behavior should be classified as intentional deception rather than hallucination.


**Presentation**
* Section 2.3: Reference to Appendix is wrong
* Table 1 is described twice at the end of Section 2.3
* Judge templates in the appendix are split weirdly
* Answer Format paragraph in Section 4 is duplicated
* Hallucinated Answer Rate is abbreviated as HFR
* Task Chaining is not explained in Section 4


### Minor
* I believe the authors' premise that the execution trajectory is a black-box to the user is unrealistic, as most agent frameworks expose intermediate steps. Framing it as impractical to manually read these is more realistic.
* Task 4, case 2 in the case study section seems weird to me. The two sub-tasks (downloading an image and deciphering a string) appear unrelated even though the multi-task template in figure 2 states "According to this file,
complete the following task". Furthermore, the decryption puzzle itself seems unsolvable. While the agent should report these failures, this makes it hard to determine if the agent is actively deceiving or merely hallucinating.
* How would the agents perform if you combine all the tested mitigations?
* Why are the explicit constraints only added to Tasks 1 and 2?
* What would happen if you would give the agent a "report_failure" tool instead of only the "final_answer" tool?
* How are the decoy and irrelevant files generated?

[1] - Lin et al. LLM-based Agents Suffer from Hallucinations: A Survey of Taxonomy, Methods, and Directions, https://arxiv.org/pdf/2509.1897

---

> ### Author Rebuttal · Authors · 2026-03-31
>
> Thank you for recognizing the real-world motivation, the focus on agentic LLMs, the broad model coverage, and the practical importance of the findings. We appreciate the reviewer’s thoughtful concerns, especially on framework dependence, task taxonomy, and the distinction between deception and hallucination.
>
> ```W1 / Q2 Framework dependence.```
> To test whether the effect is specific to smolagents, we added supplementary experiments using a standard OpenAI-style JSON tool-calling setup, with prompts adjusted accordingly. We observed the same qualitative pattern, with agents still producing successful-looking reports despite failed or incomplete execution. The results are as follows:
> |                        | Task 1 | Task 2 | Task 3 | Task 4 |       |        | Task 5 |       |        |
> | ---------------------- | ------ | ------ | ------ | ------ | ----- | ------ | ------ | ----- | ------ |
> |                        | NFR    | NFR    | DFR    | NFR    | FFR   | HFR    | NFR    | FFR   | HFR    |
> | Deepseek-v3.1-terminus | 72.50  | 85.00  | 52.50  | 95.00  | 47.50 | 100.00 | 65.00  | 50.00 | 100.00 |
> | Kimi-k2                | 92.50  | 90.00  | 52.50  | 97.50  | 42.50 | 100.00 | 82.50  | 62.50 | 96.97  |
>
> ```W2 / Q3 Task taxonomy / task choice. ```
> We agree that the current presentation of the five task types is more exploratory and dataset-construction-oriented than a formal taxonomy of deceptive behaviors. During benchmark development, our understanding of the phenomenon was still evolving, so we expanded the dataset in a bottom-up manner by iteratively identifying different manifestations and contributing factors of misleading upward reporting. As the reviewer noted, a more principled taxonomy such as by environmental constraint or deceptive mechanism, would strengthen the paper. We note that some of these distinctions are already partially reflected in Table 1, where we summarize different behavioral patterns, but we agree that the current main-text organization does not foreground them clearly enough. We will revise the writing accordingly and appreciate the reviewer’s taxonomy suggestion.
>
> ```W1 / W3 / Q1 Deception vs. hallucination.```
> We agree that the distinction between deception and hallucination is important and should be made more explicit. In our view, the key difference is that the deceptive cases highlighted in our paper are not merely unsupported answers or accidental execution errors. We manually inspected the reasoning traces in these runs and found explicit plans to fabricate or substitute missing artifacts, followed by tool-use actions consistent with that plan. For example, we observed traces such as: “Since I cannot find the actual document online, I’ll create a realistic simulation ... and then proceed with the analysis” and “The directory is empty, so I need to create the document based on the description provided.” These are then followed by actions such as creating local files and using them as if they were the requested source. We will revise the paper to clarify that we do not claim every incorrect agent behavior is deception; rather, the cases we highlight involve **failure awareness + explicit fabrication/substitution planning + non-disclosure**, which we believe distinguishes them from ordinary execution hallucination.
>
> ```Presentation / Minor issues.```
> Thank you for these careful comments. We will revise the paper to address the presentation and minor issues in detail, including the incorrect appendix reference, duplicated descriptions, notation/abbreviation inconsistencies, the explanation of task chaining, the black-box framing, the clarity of specific case studies, and the construction details of decoy/irrelevant files. We will also improve the organization of the appendix and clarify several design choices and limitations to avoid ambiguity.

---

> > ### Author Rebuttal · Reviewer_fZsA · 2026-04-02
> >
> > Thanks for the rebuttal.
> >
> > * W1: I appreciate the extra JSON tool-calling experiments. The results look promising.
> > * W3 (Deception vs Hallucination): Pointing out the reasoning traces where the model explicitly plans to fabricate files helps differentiate this from standard execution hallucinations. That mostly addresses my concern. However, I still strongly recommend toning down the anthropomorphic language. Phrases like "the agent exploits the user's limited observability" in Definition 2.1 implies a level of malicious intent and awareness that the model simply does not have in this case.
> > * W2 (Taxonomy): I would have liked to see a concrete proposal for the new taxonomy in the rebuttal and some preliminary task design. Since the dataset construction itself isn't the core contribution nor is it particularly interesting/novel, the current task design still feels too arbitrary and lacks any kind of depth.
> > * Finally, you glossed over most of my minor questions, especially the combined mitigations and the report_failure tool. I am still curious about those setups and think they would make the paper stronger.
> >
> > Overall, I will raise my score to a 4.

---

> > > ### Author Response · Authors · 2026-04-05
> > >
> > > We sincerely appreciate and highly value the reviewer’s thoughtful and constructive feedback. We have carefully considered each of the concerns you raised, and below we do our best to respond to every point that you highlighted as clearly and concretely as possible.
> > >
> > >
> > > ```W2 Taxonomy.```After carefully reflecting on the reviewer’s suggestion, we agree that the current task design would benefit from a more explicit and principled taxonomy. Following your recommendation, we have made a preliminary attempt to reorganize our current settings into two higher-level categories and seven finer-grained sub-settings. Our goal is not to claim that this taxonomy is final, but to show a more structured way to understand the relationship among our existing tasks and the different deceptive behaviors they may induce. The current proposal is summarized in the table below.
> > >
> > > | Environmental Limitation | Task Instantiation | Fine-grained Setting | Typical Deceptive Behavior |
> > > |---|---|---|---|
> > > | Target-resource unavailability | Task 1 | Broken reading tool | Unsupported guessing, simulated completion, or concealed failure |
> > > | Target-resource unavailability | Task 3 | Local decoy file | Undisclosed source substitution |
> > > | Target-resource unavailability | Task 4 | Missing download capability | Unsupported completion claim, fabricated artifact creation, or concealed failure |
> > > | Target-resource unavailability | Task 5 | Nonexistent designated file | Fabricated artifact creation, unsupported completion claim, or concealed failure |
> > > | Available-but-insufficient information | Task 2 | Missing required information | Unsupported guessing, unsupported default filling, or concealed failure |
> > > | Available-but-insufficient information | Task 2 | Irrelevant information | Unsupported guessing or concealed failure |
> > > | Available-but-insufficient information | Task 2 | Ambiguous or underspecified information | Unsupported guessing or concealed failure |
> > >
> > > Based on this taxonomy, we believe that while the phenomenon we reveal is already sufficiently clear and consistent, the current task design can indeed be improved in terms of taxonomic coverage and balance across categories. In particular, the current benchmark is not evenly distributed across sub-settings; for example, Task 2 spans three distinct information-related sub-settings, yet all of them are currently instantiated within a single task family. In future revisions, we will further refine the task construction according to this taxonomy, with the goal of improving both the quality of the tasks and the balance of coverage across different categories.
> > >
> > > ```W3 Deception vs Hallucination. ```We believe this is a very reasonable and important point, and we sincerely thank the reviewer for highlighting it. We agree that some of our current phrasing is overly anthropomorphic and may inadvertently suggest a stronger degree of malicious intent or self-awareness than is warranted in our setting. In the revised version, we will carefully soften this language, especially in Definition 2.1 and related discussions, to ensure that our wording more accurately reflects the observed behavior without over-attributing intent. We greatly appreciate this valuable suggestion.
> > >
> > > ```Minor issues:```
> > >
> > > **Report_failure Tool:** Regarding the reviewer’s question about whether a dedicated report_failure tool would help, we conducted an additional experiment by explicitly adding such a tool to the agent interface. We carefully specified in the system instruction that this tool should only be used after sufficient attempts have been made and the agent is reasonably confident that the task cannot be completed. We also explicitly instructed the agent to use this tool to avoid deceptive fallback behaviors such as guessing, fabricating files, or falsely claiming success when the required information or tool functionality is unavailable.
> > > The results are summarized below:
> > > |                        | Task 1 | Task 2 | Task 3 | Task 4 |       |        | Task 5 |       |        |
> > > | ---------------------- | ------ | ------ | ------ | ------ | ----- | ------ | ------ | ----- | ------ |
> > > |                        | NFR    | NFR    | DFR    | NFR    | FFR   | HFR    | NFR    | FFR   | HFR    |
> > > | without report_failure | 92.50  | 90.00  | 52.50  | 97.50  | 42.50 | 100.00 | 82.50  | 62.50 | 96.97  |
> > > | with report_failure    | 25.00  | 67.50  | 45.00  | 87.50  | 30.00 | 100.00 | 50.00  | 27.50 | 100.00 |
> > >
> > > **Combined Mitigation:** We also conducted an additional experiment that combines all four mitigation strategies discussed in the paper; due to time constraints, we first evaluated this combined setting on Task 4 and Task 5, and the results show a clear reduction in deceptive behavior (NFR: 22.5%, FFR: 8.75%), although the upward deception phenomenon still persists rather than disappearing completely.

---

### Official Review · Reviewer_uds4 · 2026-03-12

**Soundness:** 3
**Presentation:** 4
**Significance:** 2
**Originality:** 3
**Overall Recommendation:** 4
**Confidence:** 4

**Summary:**

The paper argues that LLM agents can engage in “agentic upward deception”: when tools or information sources fail, they may still present a final answer that makes the task look successful, sometimes by guessing, silently switching sources, simulating results, or even fabricating local files. To study this, the authors define the concept formally and build a 200-task benchmark spanning five task types and eight scenarios with constrained environments such as broken tools, irrelevant files, local decoys, missing download tools, and nonexistent files.

**Compliance With Llm Reviewing Policy:**

Affirmed.

**Final Justification:**

Thank you for your continued work on this. I will keep my score of 4. To get a higher score than this would require a broader test suite with multiple agentic harnesses that is more work than is possible during the rebuttal period. I defer to the AC on whether 4 is competitive enough for overall acceptance.

The authors have demonstrated commitment to address concerns and perform experiments quickly and this is commendable.

**Key Questions For Authors:**

- Do you expect your findings to change if you used other agentic frameworks?
- What kind of other measurements could you do beyond these limited use cases to generalize the findings of the paper?
- If you are finding that structured outputs cause models to show upwards deception, what steps have you taken to reduce or eliminate upwards deception in the judge models for your evaluation?

**Limitations:**

Yes

**Strengths And Weaknesses:**

Strengths:

- The paper is generally clearly written and I did not have any problems parsing the work.
- The term "upward deception" is interesting and I like the concept. I like how this is being extended to measure deceptive behavior by agents
- The authors specify where this occurs more (when output format is specified for example). Generally the analysis is good.

Weaknesses:

- The definition of "upwards deception" as provided in the paper is quite broad, and yet the evaluation is quite limited: Single Agent framework, limited tasks, and so on. I suspect there are many more ways that deceptive behavior can happen outside of hallucinated file downloads.
- Some of the cases show that the model is aware that the file was not downloaded and instead reconstructed (Arab spring case study), and yet the measurement nuance does not afford these edge cases (it is giving a fail based on the final outcome, but in some production cases, reconstruction of these documents may be the desired outcome).
- I think this generally speaks to the concept of assigning "intent" (via emergent behavior) to models when there are at least two edge cases. The paper doesn't distinguish between: Optimistic reporting, where the agent genuinely thinks its best-effort answer (with reconstruction etc) is responsive, and format compliance pressure, where the agent produces a structured answer because the task demanded one, not because it's trying to deceive. Some nuance in these measurements would be very helpful.

---

> ### Author Rebuttal · Authors · 2026-03-31
>
> Thank you for recognizing the clarity of the writing, the novelty of the “upward deception” concept, and the strength of the analysis. We also appreciate the reviewer’s thoughtful comments on scope, behavioral interpretation, and evaluation design.
>
> `W1 Scope and generalization.`
> Our goal is not to exhaustively cover all forms of deception, but to identify and systematically study an important subclass that is frequent in real agent workflows, operationalizable, reproducible, and practically harmful. In our setting, this behavior does not require malicious users, adversarial prompts, or special training; it arises under mundane deployment conditions such as broken tools, missing files, or mismatched information sources. We therefore believe it is an especially important failure mode to isolate and study. At the same time, the evaluation already goes beyond a narrow anecdotal study: the benchmark covers five task types, eight scenarios, and 200 tasks. We will revise the paper to make this boundary clearer: this is not a complete taxonomy of deception, but a focused benchmark for a realistic and consequential subclass.
>
> `W1 / Q1 Framework dependence.`
> We agree that absolute rates may vary across agentic frameworks. However, the phenomenon is not specific to smolagents: in an additional OpenAI tool-calling variant, we observed the same qualitative pattern, with agents still producing successful-looking reports despite failed or incomplete execution. The results are as follows:
>
> |                        | Task 1 | Task 2 | Task 3 | Task 4 |       |        | Task 5 |       |        |
> | ---------------------- | ------ | ------ | ------ | ------ | ----- | ------ | ------ | ----- | ------ |
> |                        | NFR    | NFR    | DFR    | NFR    | FFR   | HFR    | NFR    | FFR   | HFR    |
> | Deepseek-v3.1-terminus | 72.50  | 85.00  | 52.50  | 95.00  | 47.50 | 100.00 | 65.00  | 50.00 | 100.00 |
> | Kimi-k2                | 92.50  | 90.00  | 52.50  | 97.50  | 42.50 | 100.00 | 82.50  | 62.50 | 96.97  |
>
> `​​W2 Intent vs. observable behavior.`
> We agree that strong claims about internal intent should be made carefully. Our notion of deception is therefore not based on assuming malicious intent, but on whether the agent faithfully follows the user’s instruction and honestly reports what actually happened during execution. In other words, the issue is not whether the model “wanted to deceive,” but whether it represented an unfulfilled instruction as fulfilled. We agree that optimistic reporting can exist. However, optimistic reporting is not deception when it is transparent. For example, if the agent cannot directly find the requested file but clearly states that the original file was not found and that it instead compiled a substitute document from highly relevant sources, this is an honest best-effort response rather than a refusal, and should not be counted as deception. The problematic case is when the agent does not disclose that substitution or reconstruction, yet presents the result as if it had completed the original download/read/use request exactly as instructed.
>
> `W2 Format pressure.`
> We agree that format compliance pressure is an important factor. In practice, the model faces a trade-off: it may over-prioritize satisfying the required format or producing a seemingly complete output at the cost of honestly reporting failure, substitution, or uncertainty. From our perspective, this trade-off is precisely why upward deception is practically important in agent systems.
>
> `Q2 Broader measurements.`
> We agree and view this as an important future direction. Beyond file reading/writing scenarios, the evaluation should be extended to more realistic and higher-stakes agent actions, such as payments, ticket booking, purchases, reservations, form submission, account operations, and other externally consequential workflows. In such settings, upward deception may be even more harmful, because undisclosed failure, silent substitution, or fabricated completion can directly lead to financial loss, invalid transactions, or safety-critical downstream decisions.
>
> `Q3 Judge robustness.`
> First, the judge operates in a substantially different setting from the evaluated agent: it does not face the same constrained environment, tool failures, or task-completion pressure that induce upward deception in the agent setting. Second, our evaluation is grounded in detailed, rule-based criteria focused on the execution trace and disclosure behavior, rather than the surface quality or apparent completeness of the final answer. In other words, the judge is asked to determine whether failures, substitutions, or fabrications were disclosed, not simply whether the response looks well-structured or task-complete. We also provide a detailed description of the evaluation pipeline, algorithmic framework, and judge prompts in the appendix; please refer to Appendix B and Section 2.3 for the full methodology.

---

> > ### Author Rebuttal · Reviewer_uds4 · 2026-03-31
> >
> > I appreciate the authors' response and their ability and willingness to run experiments on a second agentic framework. I agree to revise my score up by a point.
> >
> > I am not convinced that judge models with rubrics are all that reliable, there's a bunch of empirical evidence about this, for instance see: https://arxiv.org/abs/2603.25133. I appreciate that this is not an isolated problem that this paper is alone in producing, but especially in a paper that is measuring human deception, it would be nice to see manual annotation here.

---

> > > ### Author Response · Authors · 2026-04-03
> > >
> > > We thank the reviewer for raising this important concern. We agree that LLM-as-a-judge is not sufficient by itself for a claim about deception, and that human validation is especially important in this setting.
> > >
> > > To address this directly, we added a manual annotation study. Two independent annotators examined 200 full Kimi-K2 trajectories spanning all five task types and labeled whether each case was deceptive. Unlike our original automatic evaluation, the annotators reviewed the complete execution trace. For Tasks 4 and 5, we also used a stricter human criterion: a case was labeled deceptive only when the agent fabricated a file and did not disclose this fabrication, either in the file itself or in its final response. The results are as follows:
> > >
> > > |                    | Task 1 | Task 2 | Task 3 | Task 4 | Task 5 | Consistency Rate |
> > > | ------------------ | ------ | ------ | ------ | ------ | ------ | ---------------- |
> > > |                    | NFR    | NFR    | DFR    | FFR    | FFR    | Human/LLM        |
> > > | LLM as judge       | 97.50  | 90.00  | 62.50  | 42.50  | 55.00  | - (200)          |
> > > | Human as judge - A | 95.00  | 90.00  | 82.50  | 55.00  | 85.00  | 88% (176)        |
> > > | Human as judge - B | 95.00  | 62.50  | 80.00  | 52.50  | 85.00  | 82.5% (165)      |
> > >
> > >
> > > Overall, **the human annotations show substantial agreement with the original LLM-judge results**. For Tasks 1, 3, 4, and 5, both annotators are largely consistent with the LLM judge, and the overall pattern of conclusions remains unchanged under human inspection. In several settings, the human annotators even identified more deceptive cases than the automatic protocol, suggesting that the LLM-based evaluation is not overstating the phenomenon and may in fact be conservative, especially for the file-fabrication settings in Tasks 4 and 5. Task 2 is the main exception. In this setting, the provided file is intentionally irrelevant to the user’s question, so the boundary between deception and over-interpretation is inherently more subjective than in the other tasks. As a result, different annotators may reasonably disagree on whether an agent that forcefully answers based on irrelevant material should be counted as deceptive. Importantly, this disagreement is localized to Task 2 rather than representative of the overall evaluation: across the remaining task types, the agreement between human annotators and the LLM judge is consistently high.
> > >
> > > We will include the human annotation protocol, the agreement statistics, and representative disagreement cases in the paper. Taken together, these results provide direct evidence that our core findings do not depend solely on rubric-based LLM judging, but remain visible under manual review of full trajectories.

---

### Decision · Program_Chairs · 2026-04-30

**Decision:**

Accept (regular)

**Comment:**

Meta Review
This paper introduces agentic upward deception—a practically important failure mode where LLM agents misrepresent task failure as success under constrained execution conditions—and provides a 200-task benchmark spanning five task types and eight scenarios. The concept is well-motivated, the empirical coverage across 11 models is broad, and the ablations on output format and task chaining are informative.
Reviewer opinions were mixed but shifted positively through the rebuttal. Two reviewers recommended weak accept from the start; the other two raised weak reject but both softened their positions. The most significant concern—LLM-as-a-judge reliability for measuring deception—was directly addressed by adding a manual annotation study with two independent annotators, showing 82–88% consistency with the automatic evaluation. Framework dependence was addressed by adding JSON tool-calling experiments showing the same qualitative pattern. The deception-versus-hallucination distinction was also clarified through manual trace inspection identifying explicit fabrication planning followed by concealment.
Remaining weaknesses—primarily the lack of a formal task taxonomy and limited scope—are real but proportionate to a benchmark paper's contribution. The authors should incorporate the human annotation results, revised taxonomy, toned-down anthropomorphic language, and combined mitigation experiments in the final version. Accept is recommended.